# Electron/infrared-phonon coupling in ABC trilayer graphene

Xiaozhou Zan [1,2,7], Xiangdong Guo [3,4,7], Aolin Deng [5], Zhiheng Huang[1,2], Le Liu[1,2], Fanfan Wu[1,2], Yalong Yuan[1,2], Jiaojiao Zhao[1,2], Yalin Peng[1,2], Lu Li[1,2], Yangkun Zhang[1,2], Xiuzhen Li[1,2], Jundong Zhu[1,2], Jingwei Dong[1,2], Dongxia Shi[1,2,6], Wei Yang [1,2,6], Xiaoxia Yang [3,4], Zhiwen Shi [5], Luojun Du [1,2] ✉, Qing Dai [3,4] ✉ & Guangyu Zhang [1,2,6] ✉

Stacking order plays a crucial role in determining the crystal symmetry and has significant impacts on electronic, optical, magnetic, and topological properties. Electron-phonon coupling, which is central to a wide range of intriguing quantum phenomena, is expected to be intricately connected with stacking order. Understanding the stacking order-dependent electron-phonon coupling is essential for understanding peculiar physical phenomena associated with electron-phonon coupling, such as superconductivity and charge density waves. In this study, we investigate the effect of stacking order on electron-infrared phonon coupling in graphene trilayers. By using gate-tunable Raman spectroscopy and excitation frequency-dependent near-field infrared nanoscopy, we show that rhombohedral ABC-stacked trilayer graphene has a significant electron-infrared phonon coupling strength. Our findings provide novel insights into the superconductivity and other fundamental physical properties of rhombohedral ABC-stacked trilayer graphene, and can enable nondestructive and high-throughput imaging of trilayer graphene stacking order using Raman scattering.

Stacking order is a unique structural degree of freedom presented in two-dimensional (2D) layered materials. It plays a significant role in governing symmetry breaking and a wide range of fascinating electronic, optical, magnetic, and topological phenomena. For instance, 3R-stacked transition metal dichalcogenides (TMDs) lack spatial inversion symmetry and exhibit intriguing valley physics and second-order nonlinear responses such as electric-dipole-allowed second harmonic generation (SHG). In contrast, inversion symmetry is restored in 2H-stacked TMD bilayers, resulting in nil valley and even-order nonlinear responses[1,2]. ABA-stacked (Bernal) and ABC-stacked

(rhombohedral) trilayer graphene have distinct electronic and optical properties[3–10]. ABC-stacked trilayer graphene exhibits an electric-field-tunable band gap and features van Hove singularities at or near the band edge, where the density of states diverges. It can also exhibit tunable Mott insulator[11], superconductor[12,13], and ferromagnetic[14,15] behavior, while these phenomena have not been observed in ABA-stacked trilayer graphene[16,17].

Electron–phonon coupling is a fundamental interaction between elementary excitations that plays a significant role in a variety of physical phenomena and quantum phase transitions in condensed

[1]Beijing National Laboratory for Condensed Matter Physics and Institute of Physics, Chinese Academy of Sciences, 100190 Beijing, China. [2]School of Physical Sciences, University of Chinese Academy of Sciences, 100190 Beijing, China. [3]CAS Key Laboratory of Nanophotonic Materials and Devices, CAS Key Laboratory of Standardization and Measurement for Nanotechnology, CAS Center for Excellence in Nanoscience, National Center for Nanoscience and Technology, 100190 Beijing, China. [4]Center of Materials Science and Optoelectronics Engineering, University of Chinese Academy of Sciences, 100049 Beijing, China. [5]Key Laboratory of Artificial Structures and Quantum Control (Ministry of Education), Shenyang National Laboratory for Materials Science, School of Physics and Astronomy, Shanghai Jiao Tong University, 200240 Shanghai, China. [6]Songshan Lake Materials Laboratory, Dongguan, 523808 Guangdong, China. [7]These authors contributed equally: Xiaozhou Zan, Xiangdong Guo. ✉e-mail: luojun.du@iphy.ac.cn; daiq@nanoctr.cn; gyzhang@iphy.ac.cn

matter physics. For example, electron–phonon coupling sets the intrinsic limit of electron mobility and leads to ultra-low thermal conductivity[18–21]. It also lays a foundation for charge density waves[22,23], electron hydrodynamics[24–26], superfluidity[27,28], and superconductivity[29–31]. As both the electronic band structures and phonon dispersions depend strongly on stacking geometry, a stacking order-governed electron–phonon coupling is expected in principle. Understanding the dependence of electron–phonon coupling on stacking order is fundamentally important for both comprehending a rich variety of peculiar physical phenomena and engineering novel device applications.

In this study, we report the observation of stacking order-governed electron–infrared phonon coupling in trilayer graphene. Specifically, we demonstrate that ABC-stacked trilayer graphene has strong electron–infrared phonon coupling strength. Our findings are supported by gate-tunable Raman spectroscopy and excitation frequency-dependent near-field infrared spectroscopy. The observed giant electron–infrared phonon coupling in ABC-stacked trilayer graphene sheds new light on its peculiar quantum properties, such as strong correlation, superconductivity, and ferromagnetism. Moreover, the distinct electron–infrared phonon coupling between ABC-stacked and ABA-stacked trilayer graphene offers a nondestructive and high-throughput imaging approach based on Raman scattering for identifying the stacking order of trilayer graphene.

## The stacking order of trilayer graphene

Trilayer graphene possesses two distinct stacking geometries that have different symmetries and electronic properties. The ABC trilayer graphene is centrosymmetric and semiconducting, while the ABA trilayer graphene is non-centrosymmetric and semimetallic (Fig. 1a and b). This unique characteristic of trilayer graphene makes it an exciting platform for studying stacking order-driven electron–phonon coupling. Both ABC and ABA trilayer graphene can be directly obtained by mechanical exfoliation from bulk crystals as they belong to structurally metastable/stable carbon allotropes. A representative trilayer graphene region with an area >5000 μm² (highlighted by the dashed black line) on 285 nm-SiO₂/Si substrates can be seen in the white-light microscopic image (Fig. 1c). However, having the same optical contrast, the ABC and ABA trilayer graphene cannot be distinguished from the optical image. Scanning near-field optical microscope (SNOM), on the other hand, can sensitively probe the electronic band structure and optical conductivity of samples, offering an effective technique to resolve the stacking order of graphene[32–35]. Figure 1d shows a typical SNOM image taken from a trilayer graphene with an excitation frequency of 940 cm⁻¹. From it, the ABC and ABA trilayer graphene can be clearly distinguished as dark and bright regions, respectively.

## Strong electron–infrared phonon coupling in the ABC trilayer graphene

Due to the interlayer coupling and stacking geometry, high-frequency optical phonon modes in graphene couple with different valence and conduction bands, resulting in different electron–phonon interaction strengths. The ABA and ABC trilayer graphene have different point group symmetries $D_{3h}$ and $D_{3d}$, respectively, corresponding to different mode symmetries: $2E' + E''$ for ABA trilayer graphene, and $2E_g + E_u$ for the ABC trilayer graphene, as shown in Fig. 2a. The $E'$ mode is both Raman and infrared active, the $E''$ and $E_g$ modes are Raman active, the antisymmetric vibrational mode $E_u$ is infrared active, and can be simultaneously Raman active through inversion symmetry breaking[36–39]. Figure 2b shows typical Raman spectra including the G mode and 2D mode of ABC and ABA trilayer graphene at heavy hole doping concentrations (ABC at P⁺⁺-red line, ABA at P⁺⁺-blue line) and charge neutral points (ABC at CNP-orange line, ABA at CNP-black line), excited by 532 nm (2.33 eV) laser at a temperature of 10 K. At heavy hole doping concentrations, the ABA trilayer graphene shows a single

Raman peak at ~1588 cm⁻¹, assigned as Raman phonon G mode. In striking contrast, ABC trilayer graphene clearly shows an additional low wave number peak at ~1572 cm⁻¹, which is absent in ABA trilayer graphene. This characteristic Raman peak is also confirmed by measurements under 633 nm-laser excitation and at 300 K (refer to Fig. S1). Besides, the 2D modes arising from a double-resonant electronic process of ABC and ABA trilayer graphene at P⁺⁺ have almost identical line shapes. At charge neutral points, the ABC and ABA trilayer graphene shows both a single Raman peak at ~1580 and ~1582 cm⁻¹, respectively. The unique low wave number peak of ABC trilayer graphene at CNP disappears. Moreover, the 2D modes of ABC and ABA trilayer graphene at CNP have different line shapes. The ABC trilayer graphene exhibits a more asymmetric and wider shape than the ABA trilayer graphene, with the 2D mode of ABC trilayer graphene including a sharp peak on the left and a flat shoulder on the right, consistent with previous observations[5,6]. The new Raman peak at ~1572 cm⁻¹ may stem from an infrared active phonon mode. To confirm this hypothesis, we carried out far-field infrared spectroscopy measurements, and the corresponding spectra are shown in Fig. 2c. Remarkably, the infrared absorption peak matches the low wavenumber Raman peak, suggesting that this low wavenumber Raman peak is the infrared active phonon mode.

As mentioned above, this infrared active phonon mode only appears in the ABC trilayer graphene with a broken inversion symmetry. To further quantitatively access the electron–infrared phonon coupling strength, we thus performed Raman spectra of both ABC and ABA trilayer graphene at different gate voltages ($V_g$) in Figs. S3–S4. Then we estimated the carrier density $n$ and Fermi level $E_F$ from $n = C_g(V_g - V_{cnp})/e$ and $E_F = -\text{sgn}(n)\hbar v_F\sqrt{(\pi|n|)}$, where $C_g = 115 \text{ aF}/\mu m^2$, $e$, $V_{cnp}$ and $v_F$ are the gate capacitance, the electron charge, the gate voltage corresponding to the CNP and the Fermi velocity[40,41]. Notably, the frequencies $\omega$ and full-width at half-maximum (FWHM) $\Gamma$ of phonon G mode of both ABA and ABC trilayer graphene are strongly $n$ dependent. Figure 2d presents the phonon energy extracted by the Lorentz fitting as a function of $n$ and $E_F$. With increasing the hole density, the mode of ABA trilayer graphene and high wavenumber mode of ABC trilayer graphene harden, indicating both are symmetric Raman G mode. In marked contrast, the low wavenumber component of ABC trilayer graphene softens with doping density, confirming the antisymmetric infrared active nature. In addition, $\Gamma$ of phonon G mode of both ABA and ABC trilayer graphene sharply decreases with increasing $n$ (Fig. 2e).

The observation of infrared active phonon mode in centrosymmetric ABC trilayer graphene by Raman spectroscopy is quite surprising since the parity is a conserved quantity, and thus Raman and infrared transitions are mutually exclusive. We firstly carried out circular polarization and linear polarization Raman measurements (Fig. S9), revealing that the phonon G mode and infrared active phonon mode have the same circular polarization responses; while the linear polarization responses $XX$ and $XY$ are almost identical, indicating that the G peak splitting we observed is not from the stress[42,43] and boundary[44–46]. Besides, Fig. S10 shows that the D peak at 1350 cm⁻¹ is almost invisible, indicating that the defects in the sample are rare, and the split G peak is not from the defects[47,48]. We thus assign this mode to the infrared active phonon mode caused by the dielectric environment doping, which breaks the inversion symmetry of ABC trilayer graphene. It is worth noting that the infrared active phonon disappears with hole doping below $|n| = \left|-10 \times 10^{12}\right| \text{ cm}^{-2}$, and can only be clearly seen with hole doping above $|n| = \left|-15 \times 10^{12}\right| \text{ cm}^{-2}$ (Figs. 2d and S4). In principle, for intrinsic graphene ($V_{CNP} = 0$ V) on 285-nm thick SiO₂ substrates, a gate voltage of $V_g = -210$ V needs to be applied to tune the carrier density up to $n = -15 \times 10^{12} \text{ cm}^{-2}$. Such high gate voltage is not practical as which far exceeds the breakdown limit of the SiO₂ gate dielectric. Thus, we measured two strong hole doping devices and one device that detected the CNP to normalize the carrier density $n$ and Fermi level $E_F$

based on the transport curve and the gate voltage corresponding to the same frequencies $\omega$ of phonon G mode in Figs. S2–S4.

We quantitatively calculated the electron–phonon coupling strength through the frequencies $\omega$ and full width at half maximum (FWHM) $\Gamma$ of phonon G mode. The change of G phonon energy is described by $\hbar\omega - \hbar\omega_{CNP} = \lambda\left\{|E_F| + \frac{\hbar\omega}{4}\ln\left|\frac{2|E_F|-\hbar\omega}{2|E_F|+\hbar\omega}\right|\right\}$, where $\omega_{CNP}$ is $\omega$ at CNP, $\lambda = A_{uc}D^2/2\pi\hbar\omega M v_F^2$. $A_{uc}$ is the area of the graphene unit cell, $M$ is the carbon atom mass, $v_F$ is the Fermi velocity and $D$ is the electron–phonon coupling strength[40,41]. The fits of the linear segments in Fig. 2d (dashed black lines) give electron–phonon coupling strength

$D_{ABA}\approx 7.0$ eV/A for ABA trilayer graphene and $D_{ABC\ G(E_g)}\approx 8.6$ eV/A for ABC trilayer graphene $E_g$ phonon mode. In addition, the electron–phonon coupling strength can also be calculated from $\Delta\Gamma = \frac{A_{uc}D^2}{8Mv_F^2}$, $\Delta\Gamma = \Gamma_{CNP} - \Gamma_0$, where $\Gamma_0$ is the residual linewidth from processes that are not related to Landau damping[40,41]. Figure 2e shows $\Delta\Gamma\approx 4$ cm$^{-1}$ for both ABA and ABC trilayer graphene $E_g$ phonon mode, giving $D_{ABA}\approx 8.0$ eV/A and $D_{ABC\ G(E_g)}\approx 8.0$ eV/A, consistent with the electron–phonon coupling strength calculated by frequencies $\omega$; while $\Delta\Gamma\approx 7$ cm$^{-1}$ for ABC trilayer graphene $E_u$ infrared active phonon mode gives $D_{ABC\ G(E_u)}\approx 10.6$ eV/A. Note that we did not see the infrared phonon in the Raman spectroscopy of ABA trilayer graphene, so we cannot directly compare the coupling strength of electron–infrared phonon between ABA and ABC trilayer graphene. But we can see that the electron–phonon coupling strength $D_{ABA}\approx 8.0$ eV/A and $D_{ABC\ G(E_g)}\approx 8.0$ eV/A are almost the same but smaller than $D_{ABC\ G(E_u)}\approx 10.6$ eV/A. In addition, the $D_{ABC\ G(E_u)}\approx 10.6$ eV/A is also stronger than the electron–phonon coupling strength 6.4eV/A in bilayer graphene[38].

Notably, recent studies have revealed superconductivity in ABC trilayer graphene[12,13] but not in ABA trilayer graphene[16,17]. Although superconductivity has also been observed in bilayer graphene in the latest research, it needs to be induced by either parallel magnetic fields[49] or WSe$_2$ proximity-induced spin–orbit coupling[50]. The stronger electron–infrared phonon coupling in ABC trilayer graphene may provide new insights into the superconductivity, as electron–phonon coupling may play a significant role.

## Resonance with the infrared active phonon in ABC trilayer graphene

We performed SNOM measurements using lasers of different frequencies to confirm the strong electron–infrared phonon coupling in ABC trilayer graphene. As shown in Fig. 3a, when the laser frequency is not at the phonon resonance frequency, the SNOM experiment reveals a low scattering amplitude intensity in ABC trilayer graphene compared

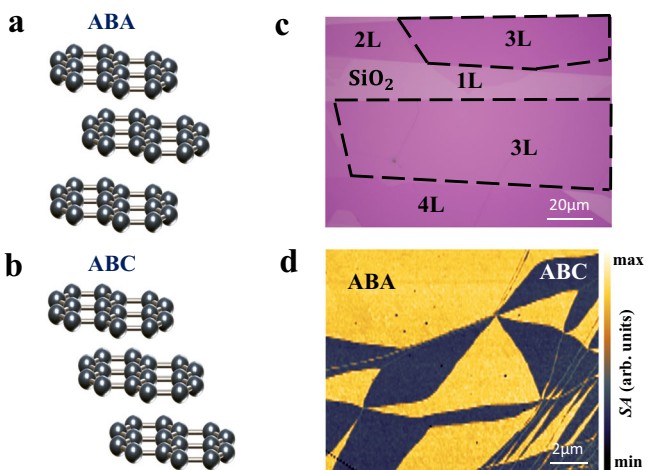

**Fig. 1 | The stacking order of trilayer graphene. a** and **b** Atomic structures of Bernal ABA-stacked and rhombohedral ABC-stacked trilayer graphene. **c** Optical microscopic image of graphene with different layers on SiO$_2$. The scale bar is 20 μm. **d** Near-field infrared nanoimaging of trilayer graphene, showing Bernal ABA (bright region) and rhombohedral ABC stacking orders (dark region). The laser frequency is 940 cm$^{-1}$. $SA$ is the near-field amplitude. The scale bar is 2 μm.

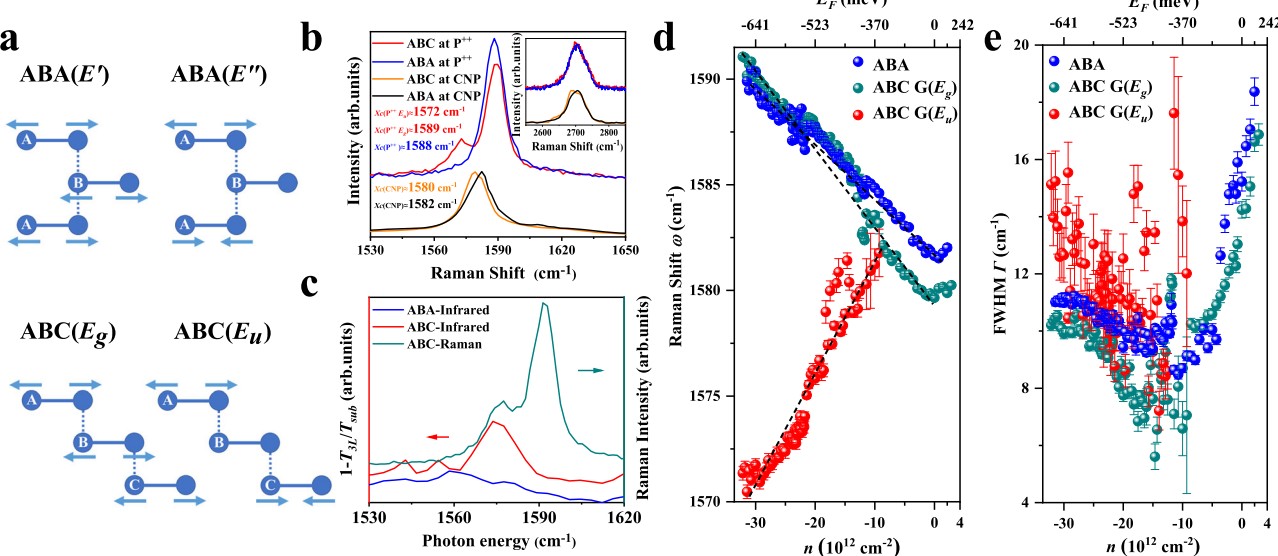

**Fig. 2 | Tunable Raman spectra of ABA and ABC trilayer graphene. a** Schematic diagram of the typical phonon vibration modes of ABA (upper panel) and ABC trilayer graphene (lower panel). **b** Raman spectra of ABA and ABC trilayer graphene at strong hole doping (ABC at P$^{++}$-red line, ABA at P$^{++}$-blue line) and charge neutral points (ABC at CNP-orange line, ABA at CNP-black line), excited by 532 nm (2.33 eV) laser at a temperature of 10 K. **c** Far-field infrared spectra of ABA (blue line) and ABC trilayer graphene (red line) are shown on the left coordinate axis, and Raman

spectrum of ABC trilayer graphene (green line) is shown on the right coordinate axis. The measurements are performed at 300 K. **d** and **e** The frequencies $\omega$ and full width at half maximum (FWHM) $\Gamma$ (by Lorentz function fitting extraction) of phonon G mode of both ABA and ABC trilayer graphene as a function of carrier density $n$ and Fermi level $E_F$. The error bar is the error range obtained by fitting the Lorentz function. The measurements are performed at 10 K.

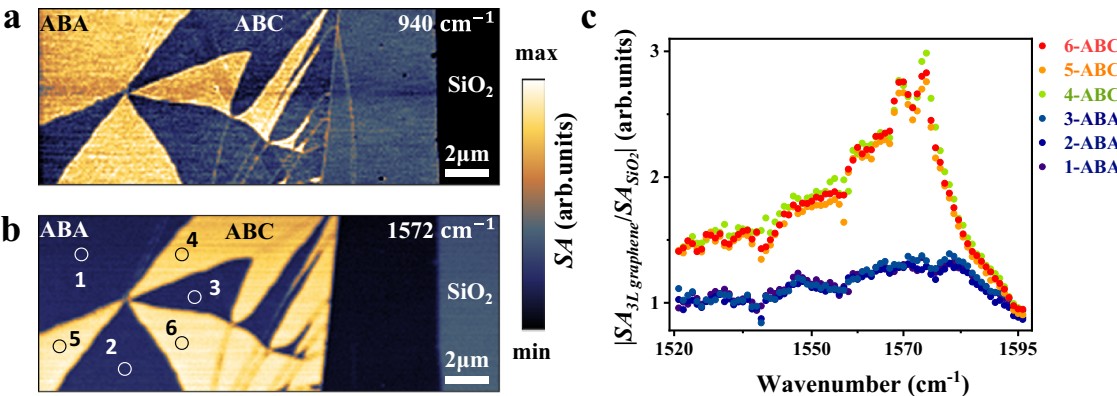

**Fig. 3 | Near-field infrared measurement of ABA and ABC trilayer graphene at different excitation frequencies. a** and **b** SNOM images were taken of the same graphene region at two different excitation frequencies: 940 and 1572 cm⁻¹. *SA* is the near-field amplitude. The scale bars are 2 μm. **c** The selected different ABA and ABC trilayer graphene regions (1–6 in **b**) for variable frequency spectral line measurements.

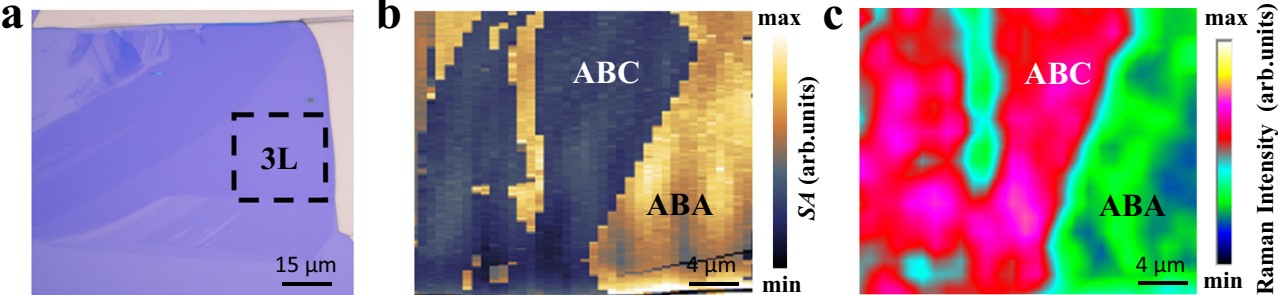

**Fig. 4 | Imaging the stacking order of trilayer graphene by Raman scattering. a** and **b** Optical microscope and SNOM diagram of trilayer graphene, the laser frequency is 940 cm⁻¹, where **b** is the area of the dotted line box in (**a**). *SA* is the near-field amplitude. The scale bars are 15 and 4 μm, respectively. **c** Imaging the stacking order of trilayer graphene by Raman scattering. The mapping Raman intensity is the Raman response of trilayer graphene at the corresponding wavenumber of infrared phonon. The scale bar is 4 μm.

to ABA trilayer graphene, indicating weak conductivity. However, when the excitation laser frequency is resonant with the infrared active phonon, the near-field signal should be dominated by the infrared phonon response if the electron–infrared phonon coupling is strong. Figure 3b shows the SNOM image excited by laser radiation resonant with the infrared active phonon at 1572 cm⁻¹ (also see Fig. S7). We performed variable frequency spectroscopy measurements (i.e., point spectroscopy measurements) on different ABA and ABC trilayer graphene regions (labeled as 1–6 in Fig. 3b). All region signal intensities were normalized using the scattering intensity of the SiO₂ substrate. We conclude that ABC has a stronger response intensity than ABA at near-infrared phonons frequencies in Fig. 3c. Electron–phonon interactions primarily arise from the coupling between phonon resonances and interband transitions, leading to the transfer of interband resonances to phonon resonances. These results suggest that the electron–infrared phonon coupling strength in ABC trilayer graphene is significantly stronger than that in ABA trilayer graphene.

## Identifying the stacking order based on the electron–infrared phonon coupling

Previously, techniques for identifying the stacking order of graphene include SNOM[32–35], SHG[51], scanning Kelvin probe microscopy (SKPM)[52], and conductive atomic force microscopy (CFM)[53,54]. Thanks to the activated infrared phonon in ABC trilayer graphene, Raman spectroscopy provides an alternative and high-throughput technique for imaging the stacking order of trilayer graphene. Figure 4a shows an optical image of mechanically exfoliated trilayer graphene. Figure 4b presents the SNOM image taken from the trilayer graphene within the dotted square area shown in Fig. 4a. ABC and ABA trilayer graphene can be clearly identified by the dark and bright regions, respectively. Figure 4c shows the Raman intensity mapping of infrared active phonon mode taken from the same region as in Fig. 4b. Obviously, the Raman mapping matches well with the SNOM mapping in terms of distinguishing the ABC and ABA regions. The Raman mapping provides a fast, simple, non-destructive, and high-throughput technology for identifying the stacking order of multilayer graphene.

In summary, our research has shown that the electron–infrared phonon coupling strength in ABC trilayer graphene is significantly stronger than in ABA trilayer graphene and AB bilayer graphene. This discovery provides new perspectives on the understanding of superconductivity and other physical properties of ABC trilayer graphene. Additionally, the differences in electron–infrared phonon coupling between ABC and ABA graphene can be utilized to develop a novel, non-destructive, and high-throughput Raman scattering imaging technique for identifying the stacking order of multilayer graphene. It also provides a new perspective for exploring and searching for new superconducting systems in graphene, such as rhombohedral multilayer graphene and twisted multilayer graphene.

## Methods
### Device fabrication
The few-layer graphene flakes were mechanically exfoliated by tape from the graphite bulk crystal and then transferred to a substrate of 285 nm-SiO₂/Si (P⁺⁺). In order to remove adsorbents on the substrate surface and ensure cleanliness of the interface, the SiO₂/Si (P⁺⁺) substrate was ultrasonically cleaned for 2 h in acetone and deionized

water, and then subjected to oxygen plasma before the process of mechanical exfoliation. Then, the transferred graphene samples were subjected to high-temperature (450 °C) annealing in an $Ar/H_2$ atmosphere for 12 h to ensure a clean surface of the graphene samples. The electrode contact was carried out by coating silver glue or evaporating Cr/Au by electron-beam metal evaporation on the edge of the graphene sample. Directly applied gate voltage through Si ($P^{++}$) substrate and used 285 nm-$SiO_2$ as the dielectric layer. The hexagonal boron nitride h-BN encapsulated graphene device is prepared through dry transfer, and a standard electron-beam lithography process and electron-beam metal evaporation.

### Raman measurement
Raman spectra are acquired using a micro-Raman spectrometer (Horiba LabRAM HR Evolution) in a confocal backscattering geometry. A solid-state laser at 532/633 nm is focused onto the samples along the $z$ direction by a ×50/×100 objective. The backscattered signal is collected by the same objective and dispersed by a 600/1800-groove $mm^{-1}$ grating. The laser power during Raman measurement is kept below 100 μW in order to avoid sample damage and excessive heating. The integration time is 10–60 s. Low-temperature (10 K) measurements were performed using a closed-cycle optical cryostat. Electrical measurements were performed in a probe station and under dark conditions with semiconductor parameter analyzers.

### Near- and far-field infrared measurement
A scattering scanning near-field optical microscope (Neaspec) equipped with a wavelength-tunable quantum cascade laser (890–2000 $cm^{-1}$) was used to image optical near fields. A metalized cantilever atomic force microscope tip served as a scattering near-field probe. The tip was illuminated with monochromatic p-polarized infrared light from a quantum cascade laser. We utilized traditional Pt metal-coated probes for the experiment. The tapping frequency of the probe was adjusted to approximately 270 kHz, while the amplitude was set to around 100 nm. Far- field infrared measurements were performed by FTIR microscopy (Thermo Fisher Nicolet iN10).

## Data availability
The data that support the findings of this study are available from the corresponding authors on a request.

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

## Acknowledgements

This work is supported by the National Key Research and Development Program of China (Grant Nos. 2023YFA1407000, 2021YFA1202900, 2020YFA0309600), National Science Foundation of China (NSFC, Grant Nos. 12274447, 61888102, 51925203, 52102160, 51972074), the Strategic Priority Research Program of CAS (Grant No. XDB0470101) and the Key-Area Research and Development Program of Guangdong Province (Grant No. 2020B0101340001).

## Author contributions

G.Z. supervised the project. X.Z. and X.G. designed the experiments. X.Z. fabricated the devices, performed the Raman spectroscopy and transport measurements with the help of Z.H. X.G., X.Y., and Q.D. carried out far-field infrared spectroscopy and SNOM variable frequency measurements. A.D. and Z.S. performed SNOM characterization measurements. X.Z. analyzed the data and prepared the figures. X.Z., X.G., L.D., and G.Z. wrote the paper. L.L., F.W., Y.Y., J.Z., Y.P., L.L., Y.Z., X.L., J.Z., J.D., W.Y., and D.S. were involved in discussions of this work.

## Competing interests

The authors declare no competing interests.
