## [Peer Review File · Nature Communications]

Electron/infrared-phonon coupling in ABC trilayer grapheneREVIEWER COMMENTS

Reviewer #1 (Remarks to the Author):

In this work, the authors reported the effect of stacking order on electron-infrared phonon coupling in trilayer graphene. They showed that the electron-infrared phonon coupling strength is much stronger in ABC-stacked trilayer graphene than in ABA-stacked ones. They observed an IR active mode at 1573 cm⁻¹ for the ABC-stacked trilayer graphene in Raman spectroscopy. This was explained by the breaking of the inversion symmetry of ABC trilayer graphene. The gate-tunable Raman spectroscopic measurements showed that the G mode of ABA- and ABC-stacked trilayer graphene blueshift with increasing hole density, but the mode at 1573 cm⁻¹ of ABC graphene redshift. Excitation energy-dependent near-field IR spectroscopy showed that the mode at 1573 cm⁻¹ is enhanced when the excitation laser is tuned to be resonant with the phonon mode due to the stronger electron-infrared phonon coupling strength. The stacking order can be probed by Raman spectroscopy due to the different electron-phonon coupling. This work showed that the electron-infrared phonon coupling can be revealed by near-field and far-field optical spectroscopy, and provides a new approach to identify the stacking order of trilayer graphene. The reviewer recommends publication of this work after minor revisions suggested below.

1. The authors showed that the mode at 1573 cm⁻¹ can be observed by 532 nm excitation at 10 K (Fig. 2b) and 633 nm at 300 K (Fig. S1) for ABC graphene. The far-field IR spectroscopy confirmed that this mode is IR active. This is an important finding. There have been several papers studying ABC-stacked graphene using Raman spectroscopy, and this mode has not been observed. In this work, is it possible that the appearance of the IR active mode may be related to the high-temperature annealing process? Did the authors measure the samples before annealing?
2. In previous work (such as Nano Lett 2011 by Tony Heinz and ACS Nano 2011 by Mildred Dresselhaus), the fwhm of the 2D band for ABC- and ABA-stacked trilayer graphene was studied and the ABC graphene showed broader 2D peaks. The analysis of the 2D band would be more informative in this work.
3. The electron-phonon coupling is temperature dependent. Did the authors compare the peak features at 10 K and 300 K?
4. In Figure S1, panels a and b are the same except that the two curves are overlapped in b. On panel would be necessary to show the difference in the two spectra.
5. Figures S4 and S5 are not discussed in the main text.

Reviewer #2 (Remarks to the Author):

The authors report SNOW and Raman study of trilayer graphene and its stacking order dependence. The main claim is the strong electron-infrared phonon coupling in the ABC-stacked trilayer graphene. Considering the recent discovery of superconductivity and other exotic properties in ABC graphene, this study can potentially provide new and useful information to advance the understanding from both experimental and theoretical aspects. The topic is of great interest, and the paper is well-written. However, a few concerns and questions need to be addressed as the main claim is not fully supported in the present form of the manuscript.

1. The main claim and conclusion of the authors is the stronger electron-infrared phonon coupling in ABC trilayer graphene. This is only supported by the presence of the Eu mode under the inversion symmetry breaking in the sample. The same mode is not available in ABA trilayer graphene, thus a direct comparison between ABC and ABA cannot be made. There is also no quantitative evidence to measure the electron-phonon coupling strength based on the formula on page 4. The authors should provide more evidence to directly support this main claim, and should also do a comparison between different systems to show the electron-phonon coupling here is indeed stronger than others. This quantitative information will also help deepen the understanding of superconductivity.

2. The trilayer graphene shown in this study is significantly far from the charge neutrality point. The authors only attribute this to the possible dielectric environment doping, but the shift is so far that the authors should do a careful study to explain what happened to this specific sample or all the samples measured.
3. The main focus of this study is the Raman active mode enabled by the broken inversion symmetry. The authors should measure ABC trilayer graphene at its charge neutrality point and confirm that this mode is absent and can be indeed induced by electrostatic doping.
4. It will be great and necessary if the authors label properly the beam spot positions on the nano-imaging of the flakes, at least in the supplementary materials.

Reviewer #3 (Remarks to the Author):

In this manuscript, Zan and coauthors report their experimental investigation of electron-phonon coupling in ABA and ABC trilayer graphene. The key experimental findings include the appearance of an infrared active phonon shown up in the Raman spectra of the centrosymmetric ABC trilayer graphene, the gate-dependent phonon frequency shift in both ABA and ABC trilayer graphene, and SNOM image contrast between on- and off-phonon resonance excitations. From these results, the authors conclude a significantly stronger electron-phonon coupling in ABC than ABA trilayer graphene. While the topic of electron-phonon coupling is an interesting topic in general and its dependence on stacking order in trilayer graphene is particularly timely, there are a few major concerns of the current version of this manuscript.

1. The electron-phonon coupling analysis in the current manuscript is too much at a qualitative level. The calculation of electron phonon coupling strength in monolayer graphene was reported in 2006-2007 [Refs: <https://doi.org/10.1103/PhysRevLett.98.166802> and <https://doi.org/10.1016/j.ssc.2007.04.022>]. The authors may adopt the formalism in both references with proper modifications to account for the trilayers to quantitatively estimate the electron-phonon coupling strength in both ABA and ABC trilayer graphene samples.
2. The G band frequency shift upon varying the gate voltage seems to be nearly the same for ABA and ABC trilayers (Figure 2e). Taking the derived equations in monolayer graphene (Equations 2 and 3 in <https://doi.org/10.1016/j.ssc.2007.04.022>), it suggests that the electron phonon coupling strength should be very similar for ABA and ABC trilayers. Can the authors please comment on this?
3. The G band linewidth change upon varying the gate voltage is another way of manifest electron-phonon coupling in graphene. Can the authors please show the linewidth dependence on the gate voltage?
4. As the authors also pointed out, this sample is heavily hole doped that prevent the show-up of Dirac point even at the highest gate voltage applied. Can the authors please use hBN encapsulation to improve the dielectric environment to resolve this problem? Having a high-quality data on a well-designed sample is crucial, especially so when targeting on a high-profile journal like Nature Communications.
5. The explanation of the appearance of the infrared active phonon mode in the Raman spectra of ABC trilayer graphene is quite speculative. And at the same time, the relationship of this infrared phonon to the electron-phonon coupling is not discussed.
6. There are in total four modes observed near the G band of ABC trilayer: 3 in the infrared spectroscopy and 2 in the Raman spectroscopy with 1 coexisting in two types of spectra. However, one would only expect 3 Davy-dov split phonons near the G band for a trilayer graphene. Can the authors comments on this discrepancy?

Point-by-point responses to Referees' comments

Reviewer #1 (Remarks to the Author):

> In this work, the authors reported the effect of stacking order on electron-infrared phonon coupling in trilayer graphene. They showed that the electron-infrared phonon coupling strength is much stronger in ABC-stacked trilayer graphene than in ABA-stacked ones. They observed an IR active mode at 1573 cm^{-1} for the ABC-stacked trilayer graphene in Raman spectroscopy. This was explained by the breaking of the inversion symmetry of ABC trilayer graphene. The gate-tunable Raman spectroscopic measurements showed that the G mode of ABA- and ABC-stacked trilayer graphene blueshift with increasing hole density, but the mode at 1573 cm^{-1} of ABC graphene redshift. Excitation energy-dependent near-field IR spectroscopy showed that the mode at 1573 cm^{-1} is enhanced when the excitation laser is tuned to be resonant with the phonon mode due to the stronger electron-infrared phonon coupling strength. The stacking order can be probed by Raman spectroscopy due to the different electron-phonon coupling. This work showed that the electron-infrared phonon coupling can be revealed by near-field and far-field optical spectroscopy, and provides a new approach to identify the stacking order of trilayer graphene. The reviewer recommends publication of this work after minor revisions suggested below.

Response 1:

We sincerely thank the Referee for recommendation of publication. We also appreciate the Referee's insightful and constructive comments for improvement. Below we address these comments in a point-by-point manner.

> 1. The authors showed that the mode at 1573 cm^{-1} can be observed by 532 nm excitation at 10 K (Fig. 2b) and 633 nm at 300 K (Fig. S1) for ABC graphene. The far-field IR spectroscopy confirmed that this mode is IR active. This is an important finding. There have been several papers studying ABC-stacked graphene using Raman spectroscopy, and this mode has not been observed. In this work, is it possible that the appearance of the IR active mode may be related to the high-temperature annealing process? Did the authors measure the samples before annealing?

Response 2:

We are delightful for the Referee's evaluation on the IR active mode in ABC graphene: "*This is an important finding*". Our experiments indicate that the observation of IR active mode requires a strong hole doping, e.g., above a carrier density of $|n| = |-15 \times 10^{12}| \text{cm}^{-2}$; and this mode is hardly seen below a carrier density of $|n| = |-10 \times 10^{12}| \text{cm}^{-2}$ (refer to Fig. R1-R3 for more data). This is the reason why this mode (the IR active mode) has not been observed previously.

Regarding the comment on "*is it possible that the appearance of the IR active mode may be related to the high-temperature annealing process?*", We have measured Raman spectra before and after annealing. In Fig. R4, we show the results from two samples (S1 and S2). Before annealing, the ABC and ABA trilayer graphene shows also both a single Raman peak at ~ 1580 and $\sim 1582\text{ cm}^{-1}$

¹, respectively; while after annealing, Raman peaks show a blue shift by 11 cm⁻¹ (sample S1) and 10 cm⁻¹ (sample S2). The IR active mode appears in ABC trilayer graphene clearly by presenting an additional low wave number peak, which is absent in ABA trilayer graphene. From these results, our answer to your question is that, yes, the appearance of the IR mode is related to the high-temperature annealing process for trilayer graphene on SiO₂, as the 450°C high temperature at Ar/H₂ atmosphere annealing process induces a strong hole doping.

To address this comment, we have included the above discussions and Fig. 2d in the revised manuscript (page 4, lines 115-116) and Fig. S2-4, 6 in the supplementary information (pages 1-2).

Figure R1. Tunable Raman spectra of ABA and ABC trilayer graphene. **a** The frequencies ω (by Lorentz function fitting extraction) of phonon G mode of both ABA and ABC trilayer graphene as a function of carrier density n and Fermi level E_F . **b** The Raman Shift of phonon G mode of both ABA and ABC trilayer graphene as a function of gate voltages. The measurements are performed at 10 K.

Figure R2. Gate-tunable Raman spectrum of ABC and ABA trilayer graphene in devices (D1, D2 and D3) with excitation wavelength of 532nm at 10K.

Figure R3. Optical microscope, SNOM images and transport curves in trilayer graphene Devices. The images from left to right are D1, D2, and D3 respectively. **a-b** Optical microscope and SNOM images. **c-d** Raman spectrum is measured at ABA and ABC point. **e** transport curves. The CNP voltage of D3 at 80V.

Figure R4. The effect of high temperature annealing in Ar/H₂ atmosphere. On the left and right of the figure are samples S1 and S2, respectively. **a-c** The Raman spectroscopy of ABC and ABA trilayer graphene after annealing (ABC -red line, ABA -blue line) and before annealing (ABC - green line, ABA -black line). This measurement is at 300K. **d-i** Optical microscope images of the sample and ABC, ABA points measured by Raman spectroscopy before and after annealing.

> 2. In previous work (such as Nano Lett 2011 by Tony Heinz and ACS Nano 2011 by Mildred Dresselhaus), the fwhm of the 2D band for ABC- and ABA-stacked trilayer graphene was studied and the ABC graphene showed broader 2D peaks. The analysis of the 2D band would be more informative in this work.

Response 3:

We appreciate the Referee's friendly suggestions. Fig. R5 illustration shows the typical Raman spectra 2D mode of ABC and ABA trilayer graphene at strong hole doping and charge neutral points. At strong hole doping, the 2D modes arising from a double-resonant electronic process of ABC and ABA trilayer graphene have almost identical line shapes. At charge neutral points, the 2D modes of ABC and ABA trilayer graphene at CNP have different line shapes. The ABC trilayer graphene exhibits a more asymmetric and wider shape than the ABA trilayer graphene, with the 2D mode of ABC trilayer graphene including a sharp peak on the left and a flat shoulder on the right. The 2D modes of ABC and ABA trilayer graphene at CNP is consistent with previous observations [Nano Lett **11**,164 (2011); ACS nano **5**, 8760(2011)].

To address this comment, we have included the above discussions and Fig.2b in the revised manuscript (page 3, lines 79-91).

Figure R5. Raman spectra of ABA and ABC trilayer graphene at strong hole doping (ABC at P⁺⁺-red line, ABA at P⁺⁺-blue line) and charge neutral points (ABC at CNP-orange line, ABA at CNP-black line), excited by 532 nm laser at a temperature of 10 K.

> 3. The electron-phonon coupling is temperature dependent. Did the authors compare the peak features at 10 K and 300 K?

Response 4:

Yes, we have compared the peak features at 10 K and 300 K, please see below Fig. R6. We extracted the Raman Shift and FWHM of phonon G mode of the ABA and ABC trilayer graphene by Lorentz function fitting. From 10K to 300K, we can see that the Raman Shift of G mode reduces 2.27 cm⁻¹ (0.48 cm⁻¹) for ABA (ABC) trilayer graphene; and the FWHM of G mode of ABA (ABC) trilayer graphene reduces 0.44 cm⁻¹ (0.63 cm⁻¹). We conclude that as the temperature increases, both the Raman Shift and the FWHM values decrease significantly.

To address this comment, we have included the above discussions and Fig.S8 in the supplementary information (pages 2, lines 59-64).

Figure R6. The Raman spectroscopy measurement at 10K and 300K. a-b The Raman spectroscopy of ABC and ABA trilayer graphene at 10K and 300K, respectively, and at the same gate voltage. **c-d** The Raman Shift and full width at half maximum (FWHM) of phonon G mode of both ABA and ABC trilayer graphene is extracted by Lorentz function fitting, respectively.

> 4. In Figure S1, panels a and b are the same except that the two curves are overlapped in b. On panel would be necessary to show the difference in the two spectra.

Response 5:

Sorry for the unclear illustration. Per referee's suggestion, we modified this figure (below Fig. R7). We can clearly see the difference in Raman spectra between ABA and ABC trilayer graphene at 633nm wavelength. The ABC trilayer graphene clearly shows an additional low wave number peak (infrared active phonon mode), which is absent in ABA trilayer graphene (Fig. R7a). Besides, the 2D modes arising from a double-resonant electronic process of ABC and ABA trilayer graphene have almost identical line shapes (Fig.R7b).

To address this comment, we have included the above discussions and Fig.S1 in the supplementary information (pages 1, lines 22-26).

Figure R7. The Raman spectrum with excitation wavelength of 633nm.

> 5. Figures S4 and S5 are not discussed in the main text.

Response 6:

Sorry for the missing discussions on Fig. S4 and S5 (the corresponding figure S9 and S10 in the revised manuscript) in the main text. We have included related discussions on them in the revised main text. Please see page 4, lines109-113 in the revised manuscript.

Reviewer #2 (Remarks to the Author):

> The authors report SNOM and Raman study of trilayer graphene and its stacking order dependence. The main claim is the strong electron-infrared phonon coupling in the ABC-stacked trilayer graphene. Considering the recent discovery of superconductivity and other exotic properties in ABC graphene, this study can potentially provide new and useful information to advance the understanding from both experimental and theoretical aspects. The topic is of great interest, and the paper is well-written. However, a few concerns and questions need to be addressed as the main claim is not fully supported in the present form of the manuscript.

Response 1:

We appreciate the positive comments from the Referee for our work “*The topic is of great interest, and the paper is well-written.*” and also appreciate the Referee’s insightful and constructive comments for improvement. We conducted in-depth and systematic measurements in the hope of addressing the Referee’s a few concerns and questions, in order to better support the main claim of our manuscript.

> 1. The main claim and conclusion of the authors is the stronger electron-infrared phonon coupling in ABC trilayer graphene. This is only supported by the presence of the Eu mode under the inversion symmetry breaking in the sample. The same mode is not available in ABA trilayer graphene, thus a direct comparison between ABC and ABA cannot be made. There is also no quantitative evidence to measure the electron-phonon coupling strength based on the formula on page 4. The authors should provide more evidence to directly support this main claim, and should also do a comparison between different systems to show the electron-phonon coupling here is indeed stronger than others. This quantitative information will also help deepen the understanding of superconductivity.

Response 2:

We greatly appreciate Referee’s suggestions and have performed systematic measurements accordingly to provide more quantitative data to support our claims and conclusions.

To quantitatively calculate the electron-infrared phonon coupling strength, we performed the Raman spectra of ABC and ABA trilayer graphene at different gate voltages (V_g) in Fig. R1-4. We convert the V_g to carrier density n and Fermi level E_F using $n = C_g(V_g - V_{CNP})/e$ and $E_F = -sgn(n)\hbar v_F\sqrt{(\pi|n|)}$, where $C_g = 115aF/\mu m^2$, e , V_{CNP} and v_F are the 285nm SiO_2 gate capacitance, the electron charge, the gate voltage corresponding to the CNP and the Fermi velocity [Phys. Rev. Lett. **98**, 166802 (2007); Solid State Communications **143**, 39 (2007)].

Note that the infrared active phonon disappears with hole doping below $|n| = |-10 \times 10^{12} cm^{-2}$ and can only be clearly seen with hole doping above $|n| = |-15 \times 10^{12} cm^{-2}$ (Fig. R1d and Fig. R4). In principle, for intrinsic graphene ($V_{CNP} = 0V$) on a substrate using 285-nm thick SiO_2 as the dielectric layer, a gate voltage of $V_g = -210V$ needs to be applied to tune the carrier density up to $n = -15 \times 10^{12} cm^{-2}$. Such high gate voltage is not practical as which far

exceeds the breakdown limit of the SiO_2 gate dielectric. Thus, we measured two strong hole doping devices and one device that detected the CNP to normalize the carrier density n and Fermi level E_F based on the transport curve and the gate voltage corresponding to the same frequencies ω of phonon G mode in Fig. R2-4.

The change of G phonon energy is: $\hbar\omega - \hbar\omega_{CNP} = \lambda \left\{ |E_F| + \frac{\hbar\omega}{4} \ln \left| \frac{2|E_F| - \hbar\omega}{2|E_F| + \hbar\omega} \right| \right\}$, where ω_{CNP} is ω at the CNP, $\lambda = A_{uc} D^2 / 2\pi\hbar\omega M v_F^2$, where A_{uc} is the area of the graphene unit cell, M is the carbon atom mass, v_F is the Fermi velocity and D is the electron-phonon coupling strength [Phys. Rev. Lett. **98**, 166802 (2007); Solid State Communications **143**, 39 (2007)]. Fitting the linear segments in Fig. R1d (dashed black lines) gives electron-phonon coupling strength $D_{ABA} \approx 7.0 eV/\text{\AA}$ for ABA trilayer graphene and $D_{ABC G(E_g)} \approx 8.6 eV/\text{\AA}$ for ABC trilayer graphene E_g phonon mode. The electron-phonon coupling strength can be calculated from $\Delta\Gamma = \frac{A_{uc} D^2}{8M v_F^2}$, $\Delta\Gamma = \Gamma_{CNP} - \Gamma_0$. Where Γ_0 is the residual linewidth from processes that are not related to Landau damping [Phys. Rev. Lett. **98**, 166802 (2007); Solid State Communications **143**, 39 (2007)]. Fig. R1e shows $\Delta\Gamma \approx 4 cm^{-1}$ for both ABA and ABC trilayer graphene E_g phonon mode, giving $D_{ABA} \approx 8.0 eV/\text{\AA}$ and $D_{ABC G(E_g)} \approx 8.0 eV/\text{\AA}$, consistent with the electron-phonon coupling strength calculated by frequencies ω of phonon G mode. While $\Delta\Gamma \approx 7 cm^{-1}$ for ABC trilayer graphene E_u infrared active phonon mode gives $D_{ABC G(E_u)} \approx 10.6 eV/\text{\AA}$.

Note that we did not see the infrared phonon in the Raman spectroscopy of ABA trilayer graphene, so we cannot directly compare the coupling strength of electron-infrared phonon between ABA and ABC trilayer graphene. But we can see that the electron-phonon coupling strength $D_{ABA} \approx 8.0 eV/\text{\AA}$ for ABA trilayer graphene and $D_{ABC G(E_g)} \approx 8.0 eV/\text{\AA}$ for ABC trilayer graphene E_g phonon mode are almost the same, they are smaller than $D_{ABC G(E_u)} \approx 10.6 eV/\text{\AA}$ for ABC trilayer graphene E_u infrared active phonon mode. In addition, the $D_{ABC G(E_u)} \approx 10.6 eV/\text{\AA}$ is also stronger than the electron-phonon coupling strength $6.4 eV/\text{\AA}$ in bilayer graphene [Phys. Rev. Lett. **101**, 257401 (2008)].

Notably, recent studies have revealed superconductivity in ABC trilayer graphene [Nature. **598**, 434 (2021); Nature **572**, 215(2019)] but not in ABA trilayer graphene [Phys. Rev. Lett. **121**, 167601 (2018); Nano Lett. **22**, 3317 (2022)]. Although superconductivity has also been observed in bilayer graphene in the latest research, it needs to be induced through parallel magnetic fields [Science **375**, 774 (2022)] or WSe₂ proximity-induced spin-orbit coupling [Nature **613**, 268(2023)]. The stronger electron-infrared phonon coupling in ABC trilayer graphene may provide new insights into its superconductivity, as electron-phonon coupling may play a significant role.

We also quantitatively measured the SNOM response intensity in ABA and ABC trilayer graphene (Fig. R5c). We performed SNOM variable frequency spectroscopy measurements (i.e., point spectroscopy measurements) on different ABA and ABC trilayer graphene regions (labeled as 1-6 in Fig. R5b). All region signal intensities were normalized using the scattering intensity of SiO_2 substrate. We

can conclude that ABC has a stronger response intensity than ABA at near infrared phonons frequencies in Fig. R5c.

To address these comments, we have included the above discussions and Fig.2-3 in the revised manuscript (page 3-5) and Fig.S2-S4 in the supplementary information (pages 1-2, lines 27-36).

Figure R1. Tunable Raman spectra of ABA and ABC trilayer graphene. **a** Schematic diagram of the typical phonon vibration modes of ABA (upper panel) and ABC trilayer graphene (lower panel). **b** Raman spectra of ABA and ABC trilayer graphene at strong hole doping (ABC at P^{++} -red line, ABA at P^{++} -blue line) and charge neutral points (ABC at CNP-orange line, ABA at CNP-black line), excited by 532 nm (2.33 eV) laser at a temperature of 10 K. **c** Far-field infrared spectra of ABA (blue line) and ABC trilayer graphene (red line) are shown on the left coordinate axis, and Raman spectrum of ABC trilayer graphene (green line) are shown on the right coordinate axis. The measurements are performed at 300K. **d-e** The frequencies ω and full width at half maximum (FWHM) Γ (by Lorentz function fitting extraction) of phonon G mode of both ABA and ABC trilayer graphene as a function of carrier density n and Fermi level E_F . The measurements are performed at 10 K.

Figure R2. Optical microscope, Snom images and transport curves in trilayer graphene Devices. The images from left to right are D1, D2, and D3 respectively. **a-b** Optical microscope and Snom images. **c-d** Raman spectrum is measured at ABA and ABC point. **c** transport curves. The CNP

voltage of D3 is at 80V.

Figure R3. The Raman Shift of phonon G mode of both ABA and ABC trilayer graphene as a function of gate voltages. The Raman Shift is extracted by Lorentz function fitting and the measurements are performed at 10 K.

Figure R4. Gate-tunable Raman spectrum of ABC and ABA trilayer graphene in devices (D1, D2 and D3) with excitation wavelength of 532nm at 10K.

Figure R5. Near-field infrared measurement of ABA and ABC trilayer graphene at different excitation frequencies. a-b SNOM images were taken of the same graphene region at two different excitation frequencies: 940 cm⁻¹ and 1572 cm⁻¹. SA is the near-field amplitude. The scales are 1 μm. **c** The selected different ABA and ABC trilayer graphene regions (1-6 in Fig.R5b) for variable frequency spectral line measurements.

> 2. The trilayer graphene shown in this study is significantly far from the charge neutrality point. The authors only attribute this to the possible dielectric environment doping, but the shift is so far that the authors should do a careful study to explain what happened to this specific sample or all the samples measured.

Response 3:

Thanks for raising these in-depth comments. We carefully reviewed the preparation process of our all graphene sample and found that the strong hole doping of the sample is caused by our annealing process. To clearly demonstrate the changes in this process, we measured Raman spectra before and after high-temperature annealing, as shown in the Fig. R6. We show the results from two samples (S1 and S2). Before annealing, the ABC and ABA trilayer graphene shows also both a single Raman peak at ~1580 and ~1582 cm⁻¹, respectively. While after annealing, Raman peaks of ABC and ABA trilayer graphene show a blue shift by 11 cm⁻¹ in the sample S1 and 10 cm⁻¹ in the sample S2. The IR active mode appears clearly in ABC trilayer graphene by presenting an additional low wave number peak, which is absent in ABA trilayer graphene. The appearance of the IR mode is related to the high-temperature annealing process for trilayer graphene on SiO₂, as the 450°C high temperature at Ar/H₂ atmosphere annealing process induces a strong hole doping.

To address this comment, we have included the above discussions and Fig.S6 in the supplementary information (pages 2, lines 48-57).

Figure R6. The effect of high temperature annealing in Ar/H₂ atmosphere. On the left and right of the figure are samples S1 and S2, respectively. **a-c** The Raman spectroscopy of ABC and ABA trilayer graphene after annealing (ABC -red line, ABA -blue line) and before annealing (ABC -green line, ABA -black line). This measurement is at 300K. **d-i** Optical microscope images of the sample and ABC, ABA points measured by Raman spectroscopy before and after annealing.

> 3. The main focus of this study is the Raman active mode enabled by the broken inversion symmetry. The authors should measure ABC trilayer graphene at its charge neutrality point and confirm that this mode is absent and can be indeed induced by electrostatic doping.

Response 4:

We appreciate the Referee's suggestions. We measured ABC trilayer graphene at its charge neutrality point and can confirm that the infrared phonon mode is absent. Fig. R1b shows the typical Raman spectra of ABC and ABA trilayer graphene at charge neutral points (ABC at CNP-orange line, ABA at CNP-black line). At charge neutral points, the ABC and ABA trilayer graphene shows both a single Raman peak at ~ 1580 and $\sim 1582 \text{ cm}^{-1}$, respectively, without any signature of the infrared phonon mode.

We can also confirm that the infrared phonon mode is observed under heavy doping conditions. The infrared active phonon disappears with hole doping below $|n| = |-10 \times 10^{12} \text{ cm}^{-2}$, and can only be clearly seen with hole doping above $|n| = |-15 \times 10^{12} \text{ cm}^{-2}$ (Fig.R1d and Fig.R4). We can confirm that the infrared active mode is absent at CNP and can be indeed induced by electrostatic doping.

To address this comment, we have included the above discussions and Fig. 2 in the revised manuscript (page 3, lines 86-91) and Fig. S2-4 in the supplementary information (pages 1, lines 27-36).

> 4. It will be great and necessary if the authors label properly the beam spot positions on the nano-imaging of the flakes, at least in the supplementary materials.

Response 5:

Thanks for these suggestions. We have provided all beam spot positions for the measurement samples, including ABA points and ABC points, as shown the Fig.R2c-d, Fig.R5c, Fig.R6f-i, d-g in the reply, Fig.3c in the main manuscript and Fig.S2c-d, Fig.S5c-d, Fig.S6f-i, d-g in the supplementary materials.

Reviewer #3 (Remarks to the Author):

> In this manuscript, Zan and coauthors report their experimental investigation of electron-phonon coupling in ABA and ABC trilayer graphene. The key experimental findings include the appearance of an infrared active phonon shown up in the Raman spectra of the centrosymmetric ABC trilayer graphene, the gate-dependent phonon frequency shift in both ABA and ABC trilayer graphene, and SNOM image contrast between on- and off-phonon resonance excitations. From these results, the authors conclude a significantly stronger electron-phonon coupling in ABC than ABA trilayer graphene. While the topic of electron-phonon coupling is an interesting topic in general and its dependence on stacking order in trilayer graphene is particularly timely, there are a few major concerns of the current version of this manuscript.

Response 1:

We sincerely thank the Referee for the positive comments and recognition of our work “*the topic of electron-phonon coupling is an interesting topic in general and its dependence on stacking order in trilayer graphene is particularly timely*” and also appreciate the Referee’s insightful and constructive comments for improvement. Below, we will discuss the comments one by one and provide more convincing data to address a few concerns.

> 1. The electron-phonon coupling analysis in the current manuscript is too much at a qualitative level. The calculation of electron phonon coupling strength in monolayer graphene was reported in 2006-2007

[Refs: <https://doi.org/10.1103/PhysRevLett.98.166802> and <https://doi.org/10.1016/j.ssc.2007.04.022>]. The authors may adopt the formalism in both references with proper modifications to account for the trilayers to quantitatively estimate the electron-phonon coupling strength in both ABA and ABC trilayer graphene samples.

Response 2:

We agree with the Referee that the quantitative analysis would be more helpful and greatly appreciate Referee's suggestions. We thus performed systematic measurements to quantitatively provide the electron-phonon coupling strength in trilayer graphene.

To quantitatively calculate the electron-infrared phonon coupling strength, we performed the Raman spectra of ABC and ABA trilayer graphene at different gate voltages (V_g) in Fig. R1-4. We convert the V_g to carrier density n and Fermi level E_F using $n = C_g(V_g - V_{CNP})/e$ and $E_F = -sgn(n)\hbar v_F\sqrt{(\pi|n|)}$, where $C_g = 115aF/\mu m^2$, e , V_{CNP} and v_F are the 285nm SiO_2 gate capacitance, the electron charge, the gate voltage corresponding to the CNP and the Fermi velocity [Phys. Rev. Lett. **98**, 166802 (2007); Solid State Communications **143**, 39 (2007)].

Note that the infrared active phonon disappears with hole doping below $|n| = |-10 \times 10^{12}|cm^{-2}$ and can only be clearly seen with hole doping above $|n| = |-15 \times 10^{12}|cm^{-2}$ (Fig. R1d and Fig. R4). In principle, for intrinsic graphene ($V_{CNP} = 0V$) on a substrate using 285-nm

thick SiO_2 as the dielectric layer, a gate voltage of $V_g = -210V$ needs to be applied to tune the carrier density up to $n = -15 \times 10^{12} cm^{-2}$. Such high gate voltage is not practical as which far exceeds the breakdown limit of the SiO_2 gate dielectric. Thus, we measured two strong hole doping devices and one device that detected the CNP to normalize the carrier density n and Fermi level E_F based on the transport curve and the gate voltage corresponding to the same frequencies ω of phonon G mode in Fig. R2-4.

The change of G phonon energy is: $\hbar\omega - \hbar\omega_{CNP} = \lambda \left\{ |E_F| + \frac{\hbar\omega}{4} \ln \left| \frac{2|E_F| - \hbar\omega}{2|E_F| + \hbar\omega} \right| \right\}$, where ω_{CNP} is ω at the CNP, $\lambda = A_{uc} D^2 / 2\pi \hbar \omega M v_F^2$, where A_{uc} is the area of the graphene unit cell, M is the carbon atom mass, v_F is the Fermi velocity and D is the electron-phonon coupling strength [Phys. Rev. Lett. **98**, 166802 (2007); Solid State Communications **143**, 39 (2007)]. Fitting the linear segments in Fig. R1d (dashed black lines) gives electron-phonon coupling strength $D_{ABA} \approx 7.0 eV/\text{\AA}$ for ABA trilayer graphene and $D_{ABC G(E_g)} \approx 8.6 eV/\text{\AA}$ for ABC trilayer graphene E_g phonon mode. The electron-phonon coupling strength can be calculated from $\Delta\Gamma = \frac{A_{uc} D^2}{8M v_F^2}$, $\Delta\Gamma = \Gamma_{CNP} - \Gamma_0$. Where Γ_0 is the residual linewidth from processes that are not related to Landau damping [Phys. Rev. Lett. **98**, 166802 (2007); Solid State Communications **143**, 39 (2007)]. Fig. R1e shows $\Delta\Gamma \approx 4 cm^{-1}$ for both ABA and ABC trilayer graphene E_g phonon mode, giving $D_{ABA} \approx 8.0 eV/\text{\AA}$ and $D_{ABC G(E_g)} \approx 8.0 eV/\text{\AA}$, consistent with the electron-phonon coupling strength calculated by frequencies ω of phonon G mode. While $\Delta\Gamma \approx 7 cm^{-1}$ for ABC trilayer graphene E_u infrared active phonon mode gives $D_{ABC G(E_u)} \approx 10.6 eV/\text{\AA}$.

Note that we did not see the infrared phonon in the Raman spectroscopy of ABA trilayer graphene, so we cannot directly compare the coupling strength of electron-infrared phonon between ABA and ABC trilayer graphene. But we can see that the electron-phonon coupling strength $D_{ABA} \approx 8.0 eV/\text{\AA}$ for ABA trilayer graphene and $D_{ABC G(E_g)} \approx 8.0 eV/\text{\AA}$ for ABC trilayer graphene E_g phonon mode are almost the same, they are smaller than $D_{ABC G(E_u)} \approx 10.6 eV/\text{\AA}$ for ABC trilayer graphene E_u infrared active phonon mode.

We also quantitatively measured the SNOM response intensity in ABA and ABC trilayer graphene (Fig. R5c). We performed SNOM variable frequency spectroscopy measurements (i.e., point spectroscopy measurements) on different ABA and ABC trilayer graphene regions (labeled as 1-6 in Fig. R5b). All region signal intensities were normalized using the scattering intensity of SiO_2 substrate. We can conclude that ABC has a stronger response intensity than ABA at near infrared phonons frequencies in Fig. R5c.

To address these comments, we have included the above discussions and Fig.2-3 in the revised manuscript (page 3-5) and Fig.S2-S4 in the supplementary information (pages 1-2, lines 27-36).

Figure R1. Tunable Raman spectra of ABA and ABC trilayer graphene. **a** Schematic diagram of the typical phonon vibration modes of ABA (upper panel) and ABC trilayer graphene (lower panel). **b** Raman spectra of ABA and ABC trilayer graphene at strong hole doping (ABC at P⁺⁺-red line, ABA at P⁺⁺-blue line) and charge neutral points (ABC at CNP-orange line, ABA at CNP-black line), excited by 532 nm (2.33 eV) laser at a temperature of 10 K. **c** Far-field infrared spectra of ABA (blue line) and ABC trilayer graphene (red line) are shown on the left coordinate axis, and Raman spectrum of ABC trilayer graphene (green line) are shown on the right coordinate axis. The measurements are performed at 300K. **d-e** The frequencies ω and full width at half maximum (FWHM) Γ (by Lorentz function fitting extraction) of phonon G mode of both ABA and ABC trilayer graphene as a function of carrier density n and Fermi level E_F . The measurements are performed at 10 K.

Figure R2. Optical microscope, Snom images and transport curves in trilayer graphene Devices. The images from left to right are D1, D2, and D3 respectively. **a-b** Optical microscope and Snom images. **c-d** Raman spectrum is measured at ABA and ABC point. **c** transport curves. The CNP voltage of D3 at 80V.

Figure R3. The Raman Shift of phonon G mode of both ABA and ABC trilayer graphene as a function of gate voltages. The Raman Shift is extracted by Lorentz function fitting and the measurements are performed at 10 K.

Figure R4. Gate-tunable Raman spectrum of ABC and ABA trilayer graphene in devices (D1, D2 and D3) with excitation wavelength of 532nm at 10K.

Figure R5. Near-field infrared measurement of ABA and ABC trilayer graphene at different excitation frequencies. a-b SNOM images were taken of the same graphene region at two different excitation frequencies: 940 cm^{-1} and 1572 cm^{-1} . SA is the near-field amplitude. The scales are 1 μm . c The selected different ABA and ABC trilayer graphene regions(1-6 in Fig.R20b) for variable frequency spectral line measurements.

> 2. The G band frequency shift upon varying the gate voltage seems to be nearly the same for ABA and ABC trilayers (Figure 2e). Taking the derived equations in monolayer graphene (Equations 2 and 3 in <https://doi.org/10.1016/j.ssc.2007.04.022>), it suggests that the electron phonon coupling strength should be very similar for ABA and ABC trilayers. Can the authors please comment on this?

Response 3:

We did not see the infrared phonon in the Raman spectroscopy of ABA trilayer graphene, so we cannot directly compare the coupling strength of electron-infrared phonon between ABA and ABC trilayer graphene. As shown in above responses, we calculate the electron-phonon coupling strength $D_{ABA} \approx 8.0 \text{ eV}/\text{\AA}$ for ABA trilayer graphene and $D_{ABCG(E_g)} \approx 8.0 \text{ eV}/\text{\AA}$ for ABC trilayer graphene E_g phonon mode are almost the same, they are smaller than $D_{ABCG(E_u)} \approx 10.6 \text{ eV}/\text{\AA}$ for ABC trilayer graphene E_u infrared active phonon mode.

> 3. The G band linewidth change upon varying the gate voltage is another way of manifest electron-phonon coupling in graphene. Can the authors please show the linewidth dependence on the gate voltage?

Response 4:

Sure, here we show the line width dependence on the gate voltage. The full width at half maximum (FWHM) Γ of phonon G mode of both ABA and ABC trilayer graphene extracted by Lorentz fitting are strongly carrier density n dependent in Fig.R1e. Γ of phonon G mode of both ABA and ABC trilayer graphene sharply decreases with increasing the hole density. In addition, the electron-phonon coupling strength can also be calculated from $\Delta\Gamma = \frac{A_{uc}D^2}{8Mv_F^2}$, $\Delta\Gamma = \Gamma_{\text{CNP}} - \Gamma_0$. where Γ_0 is the residual linewidth from processes that are not related to Landau damping [Phys. Rev.

Lett.**98**, 166802 (2007); Solid State Communications **143**, 39 (2007)]. Fig. R1e shows that the $\Delta\Gamma \approx 4\text{cm}^{-1}$ for ABA trilayer graphene calculates $D_{ABA} \approx 8.0\text{eV}/\text{\AA}$, the $\Delta\Gamma \approx 4\text{cm}^{-1}$ for ABC trilayer graphene E_g phonon mode calculates $D_{ABCG(E_g)} \approx 8.0\text{eV}/\text{\AA}$ and the $\Delta\Gamma \approx 7\text{cm}^{-1}$ for ABC trilayer graphene E_u infrared active phonon mode calculates $D_{ABCG(E_u)} \approx 10.6\text{eV}/\text{\AA}$.

To address these comments, we have included the above discussions and Fig.2 in the revised manuscript (page 4, lines 127-135)

> 4. As the authors also pointed out, this sample is heavily hole doped that prevent the show-up of Dirac point even at the highest gate voltage applied. Can the authors please use hBN encapsulation to improve the dielectric environment to resolve this problem? Having a high-quality data on a well-designed sample is crucial, especially so when targeting on a high-profile journal like Nature Communications.

Response 5:

We sincerely thank the referee for this insightful suggestion. We thus fabricated the trilayer graphene device encapsulated in hexagonal boron nitride (h-BN), as shown in Figure R6. The transport curve shows that the voltage at the CNP is $\sim 0\text{V}$ in Fig. R6b. Fig. R6 e-f show the Raman spectrum of phonon G and 2D mode at different gate voltage, respectively. At CNP, the ABC and ABA trilayer graphene shows also both a single Raman peak at ~ 1580 and $\sim 1582\text{cm}^{-1}$, respectively. Moreover, the 2D modes of ABC and ABA trilayer graphene at CNP have different line shapes. Because the new infrared active phonon disappears with hole doping below $|n| = |-10 \times 10^{12}\text{cm}^{-2}$, and can only be clearly seen with hole doping above $|n| = |-15 \times 10^{12}\text{cm}^{-2}$ (Fig.R1d and Fig.R4). We did not observe the infrared phonon mode in ABC trilayer graphene when we detected the corresponding carrier carrier density $n \approx \pm 7 \times 10^{12}\text{cm}^{-2}$ within the gate voltage range of $\pm 100\text{V}$ (CNP at 0V), which is almost approaching the breakdown limit of the dielectric layer. It is consistent with the above discussion. The is quite surprising that the new infrared active phonon requires the strong hole doping above $|n| = |-15 \times 10^{12}\text{cm}^{-2}$.

To address these comments, we have included the above discussions Fig. S5 in the supplementary information (pages 2, lines 37-47).

Figure R6. The Raman spectrum and transport curve in h-BN encapsulated trilayer graphene device. **a** The snom nanoimage of h-BN encapsulated trilayer graphene device, the thicknesses of top BN and bottom BN are $\sim 2\text{nm}$ and $\sim 25\text{nm}$. **b** transport curve. **c-d** Raman spectrum is measured at ABA and ABC point. **e-f** The phonon G and 2D mode of the Raman spectrum at different gate voltages with excitation wavelength of 532nm. The measurement is at 10K

> 5. The explanation of the appearance of the infrared active phonon mode in the Raman spectra of ABC trilayer graphene is quite speculative. And at the same time, the relationship of this infrared phonon to the electron-phonon coupling is not discussed.

Response 6:

We have provided more data to confirm that the infrared active mode is absent at CNP and can be indeed induced by electrostatic doping. And we quantitatively calculated the electron-infrared phonon coupling strength in ABC trilayer graphene.

We first measured the Raman spectra of ABC and ABA trilayer graphene at strong hole doping (ABC at P^{++} -red line, ABA at P^{++} -blue line) and charge neutral points (ABC at CNP-orange line, ABA at CNP-black line). The ABC trilayer graphene at strong hole doping clearly shows an additional low wave number peak at $\sim 1572\text{ cm}^{-1}$, which is absent in ABA trilayer graphene at strong hole doping and ABA, ABC trilayer graphene at CNP. Then Fig. R1d shows that with increasing the hole density, the mode of ABA trilayer graphene and high wavenumber mode of ABC trilayer graphene harden, indicating both are symmetric Raman G mode. In marked contrast, the low wavenumber component of ABC trilayer graphene softens with doping density, confirming the antisymmetric infrared active nature. And the infrared active phonon disappears with hole doping below $|n| = |-10 \times 10^{12}| \text{cm}^{-2}$, and can only be clearly seen with hole doping above $|n| = |-15 \times 10^{12}| \text{cm}^{-2}$ (Fig.1d and Fig. R4).

We have also carried out circular polarization and linear polarization Raman spectra measurements (Fig. R7), and it can be concluded that phonon G mode and infrared active phonon mode have the same circular polarization response, while the linear polarization response XX and XY are almost identical,

indicating that the G peak splitting we observed is not from the stress [Phys. Rev. B **79**, 205433 (2009);ACS nano. **5**, 2231 (2011)] and the boundary [Nano Lett.**11**, 4083 (2011); Phys. Rev. B **81**, 035412 (2010)]. At the same time, Fig. R8 shows that the 1350cm^{-1} D peak of the sample is almost invisible, indicating that the defects of the sample itself are few, and the separation of the G peak is not from the defects [Nano Lett.**12**, 3925(2012); Phys. Rev. B **84**, 035433 (2011)]. So, the Raman activity of infrared active phonon modes is attributed to the dielectric environment doping, which breaks the inversion symmetry of ABC trilayer graphene. We can confirm that the infrared active mode is absent at CNP and can be indeed induced by electrostatic doping.

At the same time, we have discussed in detail the relationship of this infrared phonon to the electron-phonon coupling in revised manuscript. And we quantitatively provided the electron- E_u infrared active phonon coupling strength in ABC trilayer graphene $D_{ABC G(E_u)} \approx 10.6\text{eV}/\text{\AA}$ in above Responses.

To address these comments, we have included the above discussions and Fig.2 in the revised manuscript (page 3-4, lines 79-91,102-105,109-116) and Fig.S2-4,9-10 in the supplementary information (pages 1-3, lines 27-36,64-71).

Figure R7. Circularly polarized and linearly polarized Raman spectra. a Circularly polarized Raman spectra. **b** Linearly polarized Raman spectra. They were measured at 300K.

Figure R8. Raman spectrum of 1350cm^{-1} D peak of trilayer graphene.

> 6. There are in total four modes observed near the G band of ABC trilayer: 3 in the infrared spectroscopy and 2 in the Raman spectroscopy with 1 coexisting in two types of spectra. However, one would only expect 3 Davy-dov split phonons near the G band for a trilayer graphene. Can the authors comments on this discrepancy?

Response 7:

Yes, you're right. According to the theory and first principles calculations [Phys. Rev. B **77**, 125401 (2008) and Phys. Rev. B **79**, 115443 (2009)], 3 Davy-dov split phonons near the G band in the ABC trilayer graphene are expected (Fig.R9), with corresponding linewidths of $\gamma = 7.2\text{cm}^{-1}$ of $E_{g,a}$ phonon mode, $\gamma = 0.0\text{cm}^{-1}$ of E_u phonon mode and $\gamma = 0.3\text{cm}^{-1}$ of $E_{g,b}$. The $E_{g,a}$ and $E_{g,b}$ modes are Raman active, E_u is infrared active, and the antisymmetric vibrational mode E_u can be simultaneously Raman active through inversion symmetry breaking [Phys. Rev. B **77**, 125401 (2008); J. Phys. Soc. Jpn.**78**, 034709 (2009); Phys. Rev. Lett.**101**, 257401 (2008) and Phys. Rev. B **80**, 241417 (2009)], and then the linewidth γ of E_u phonon mode will be much greater than 0. But, the linewidth $\gamma = 0.3\text{cm}^{-1}$ of $E_{g,b}$ is very small and is difficult to observe in experiments. So we can only observe two phonon modes $E_{g,a}$ and E_u in Raman spectroscopy near the G band of ABC trilayer.

According to the theory and first principles calculations [Phys. Rev. B **84**, 245433 (2011)], the anisotropy of the infrared spectra with respect to the direction of the light electric field is significant, and such as boundary, disorder, or defect effects, which may cause the splitting of the infrared peak in the infrared spectroscopy.

TABLE I. Calculated phonon linewidth γ (in cm^{-1}) for the high-frequency optical-phonon modes at Γ and K in monolayer, bilayer, and trilayer graphenes. The mode symmetries S and the frequencies ω (in cm^{-1}) are also listed for completeness.

	Monolayer			AB			ABA			ABC		
	S	ω	γ	S	ω	γ	S	ω	γ	S	ω	γ
Γ	E_{2g}	1586	11.2	E_g	1587	8.6	E'_a	1586	9.7	$E_{g,a}$	1586	7.2
				E_u	1592	0.1	E''	1588	11.0	E_u	1589	0.0
							E'_b	1593	2.8	$E_{g,b}$	1594	0.3
K	A'_1	1306	20.4	E	1318	9.0	E'_1, E''_1	1316	8.4	E	1318	2.8
							E'_2	1324	3.6	A_1	1325	2.2

Figure R9. Refer to Phys. Rev. B **79, 115443 (2009)**

REVIEWERS' COMMENTS

Reviewer #1 (Remarks to the Author):

The authors answered my questions and made corresponding changes in the revised manuscript. The revised manuscript can now be recommended for publication in Nature Communications. There is only one minor revision: In the caption of Figure 3, the scale bars are 1 μm , while in the figure, it is 2 μm .

Reviewer #2 (Remarks to the Author):

The authors have addressed my questions with additional data and discussions. The manuscript has been well improved. I would recommend the publication.

Reviewer #3 (Remarks to the Author):

In this round of revision, the authors have addressed all my questions thoroughly and satisfactorily, with new experimental data and in-depth explanations. I have no further questions and recommend the acceptance of this work at Nature Communications.

[revised manuscript text omitted]